# Variable-rate hierarchical CPC
# leads to acoustic unit discovery in speech

**Santiago Cuervo**[1,2]    **Adrian Łańcucki**[3]    **Ricard Marxer**[2]

**Paweł Rychlikowski**[1]    **Jan Chorowski**[4]

[1] University of Wrocław, Poland
[2] Université de Toulon, Aix Marseille Univ, CNRS, LIS, France
[3] NVIDIA, Poland
[4] Pathway, France

## Abstract

The success of deep learning comes from its ability to capture the hierarchical structure of data by learning high-level representations defined in terms of low-level ones. In this paper we explore self-supervised learning of hierarchical representations of speech by applying multiple levels of Contrastive Predictive Coding (CPC). We observe that simply stacking two CPC models does not yield significant improvements over single-level architectures. Inspired by the fact that speech is often described as a sequence of discrete units unevenly distributed in time, we propose a model in which the output of a low-level CPC module is non-uniformly downsampled to directly minimize the loss of a high-level CPC module. The latter is designed to also enforce a prior of separability and discreteness in its representations by enforcing dissimilarity of successive high-level representations through focused negative sampling, and by quantization of the prediction targets. Accounting for the structure of the speech signal improves upon single-level CPC features and enhances the disentanglement of the learned representations, as measured by downstream speech recognition tasks, while resulting in a meaningful segmentation of the signal that closely resembles phone boundaries.

## 1    Introduction

Perceptual signals, such as speech, images, or natural language are hierarchical, and exhibit a non-uniform information density. We believe that by caring for these two aspects of the data we can learn improved representations of such signals. Modern self-supervised representation learners do so only partially: while they are implemented using deep neural networks which internally learn a hierarchy of concepts, the usual training criteria are only applied at the top layer and do not fully benefit from the hierarchical nature of the signals. Moreover, the models often assume a uniform information density and process the data at fixed input-driven rates (e.g., by striding or uniform pooling over time or pixels on speech and images, respectively). We demonstrate that the quality of learned representations can be improved by explicitly modeling the data at two different, non-uniform sampling rates, and with different training criteria applied at each level of data modeling.

We demonstrate our results on speech, which is a hierarchical signal with a well understood structure. We extend a frame-wise feature extraction model with a boundary predictor, followed by another feature transformation applied to variable length contiguous segments of frames. We show that through a careful design of the second level training criterion we can improve on the quality of

36th Conference on Neural Information Processing Systems (NeurIPS 2022).

the learned representations and perform unit discovery by detecting boundaries which overlap with ground-truth segmentation of the data. Therefore, we show the benefits of coupling variable-rate data representations with several levels of contrastive training.

## 1.1 Background: unsupervised representation learning

The deep learning revolution was started by layerwise stacking of pre-trained shallow models such as Restricted Boltzman Machines [Hinton and Salakhutdinov, 2006] or denoising autoencoders [Vincent et al., 2008]. However, it was soon observed that deep models can be trained from scratch without the need for unsupervised pretraining [Krizhevsky et al., 2012]. Consequently, unsupervised feature modeling was limited to learning shallow word representations [Mikolov et al., 2013]. Recently, novel self-supervised training criteria such as unmasking [Devlin et al., 2019], or contrastive predictions [van den Oord et al., 2018, Baevski et al., 2020, Chen et al., 2020] have facilitated pre-training deep networks on unlabeled datasets.

Despite these advances, improving the quality of self-supervised representations of data by modeling the signal at several levels of hierarchy remains an unsolved problem. Löwe et al. [2019] have stacked several Contrastive Predictive Coding (CPC, van den Oord et al. [2018]) modules, however their goal was not to improve the representations, but to avoid a global backpropagation training. In the speech domain, their pre-training regime has failed to match the quality attained by training of a single deep CPC network. Bhati et al. [2021] introduced a hierarchical model which stacked two CPC modules separated by a differentiable boundary detector. They demonstrated that adding the second level of CPC modeling leads to improved phone boundary detection, but at the expense of the predictive capability of the learned features. This tension between boundary detection and feature informativeness was further confirmed by Cuervo et al. [2022].

We demonstrate that it is possible to improve the predictive quality of features by introducing a second CPC level which is trained on variable length segments extracted to directly optimize the performance of the second-level CPC criterion. This result has several implications. First, it demonstrates the benefits of more closely matching the variable information rate of speech. Second, it paves the way for unsupervised discovery of structure in data, for instance acoustic unit discovery in speech.

## 1.2 Unit discovery in speech

Speech is a well-suited benchmark modality for structure discovery using predictive coding methods. Patterns at the level of acoustic frames change in a predictable way due to the physical constraints of the vocal apparatus. Yet these evolve depending on higher level latent variables, such as phones or prosody. Due to the structure of language, speech is also predictable at the level of phones: in English we would expect the phone "IH" to be followed by the phone "T" to form the word "it". At word level, we can predict that the pronoun "it" will be followed by a verb like "is" or "does". The same rationale can be applied to other language-related signals such as handwritten text.

Furthermore, acoustic units have some easy to characterise properties. For instance, phone durations are limited by the rate at which they can be articulated and perceived, averaging around 100 ms in English. Useful units are also contrastive, for easier distinguishing between subsequent phones by a listener, and inherently redundant, which makes them robust to external noise, but also more predictable. Existing approaches to acoustic unit discovery impose those properties on the features extracted with a frame-level, self-supervised model. The main idea behind our method is to enforce such priors of average duration, contrastiveness and predictability in a differentiable, self-supervised objective function, which a two-level formulation permits. By optimizing this function we learn to extract information bearing units from unlabeled continuous sequential data.

## 2 Model

We describe the operation of our model in the speech domain, noting that the design considerations involved could apply to other kinds of sequential data with an inherent discrete structure, such as handwriting encoded as long lines of written characters. The major parts of the model (Figure 1) are two CPC modules, separated by a trained variable-rate downsampling segment extractor.

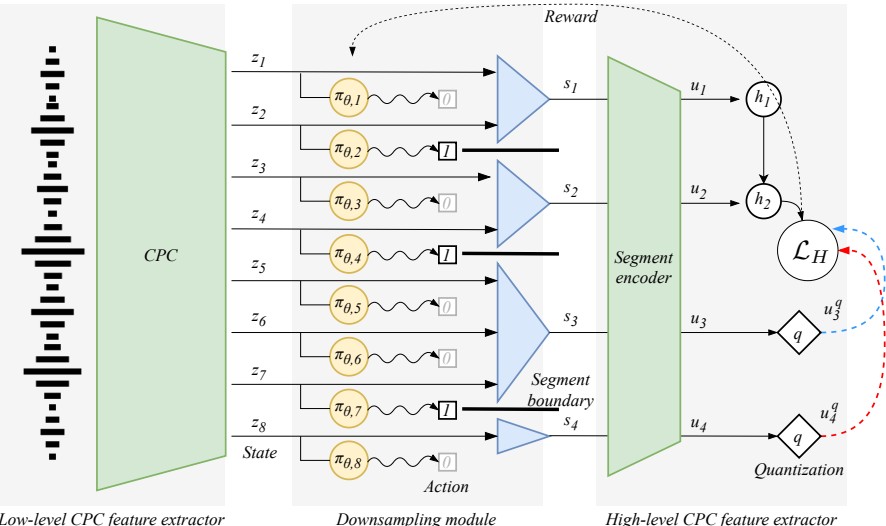

Figure 1: The model is composed of three parts. The low-level feature extractor is a CPC model which processes the signal at the frame level. The downsampling module groups and averages segments of adjacent representations. It is trained by reinforcement learning to compresses the features depending on their information content. Finally, the high-level CPC feature extractor models slowly changing features. The loss $\mathcal{L}_H$ is used as a reward to guide training of the policy. Low-level CPC is depicted as a black box, and high-level CPC in detail to show quantization and adjacent negative sampling.

**Low-level feature extractor** The feature extractor is a typical CPC network for audio data [van den Oord et al., 2018, Rivière et al., 2020] that models the regularities of the signal at the acoustic level. It takes as input the raw waveform $x$ and maps it to sequences of encodings $z \in \mathbb{R}^{T \times d}$ and context vectors $c \in \mathbb{R}^{T \times d}$, both $d$-dimensional vectors calculated for all $T$ time steps. The encodings are produced by a strided convolutional encoder $g_{enc}$ and contain local information on the receptive field of the encoder at time $t$: $z_t = g_{enc}(x_t)$. The context vectors are the output of a recurrent neural network $g_{ar}$ which summarizes observations up to time $t$: $c_t = g_{ar}(z_{<t})$.

**Downsampling module** This module aims to downsample the frame representations depending on their information content so as to improve the performance of the higher level feature extractor. It contains a boundary predictor and an average pooling layer. The boundary predictor is a transformer network with bidirectional attention [Vaswani et al., 2017] that predicts a sequence of unit boundary indicators $b \in \{0, 1\}^T$ conditioned on $z$. For two consecutive non-zero elements in $b$: $b_t$ and $b_{t+l}$, the pooling layer compresses the sub-sequence of encodings between boundaries $z_t, \ldots, z_{t+l-1}$ into a single segment representation $s = \frac{1}{l} \sum_{i=0}^{l-1} z_{t+i}$.

**High-level feature extractor** After downsampling, the resulting sequence of segment representations is processed by a second CPC network aiming to model the signal at the level of discrete units. It maps the sequence $s \in \mathbb{R}^{T' \times d'}$ to sequences of pseudo-unit representations $u \in \mathbb{R}^{T' \times d'}$ and unit-level context representations $h \in \mathbb{R}^{T' \times d'}$. The pseudo-unit representations are produced by a fully-connected network $s_{enc}$: $u_k = s_{enc}(s_t)$. The context vectors are the output of a recurrent neural network $s_{ar}$ which summarizes the history of pseudo-units up to the $k$-th (high-level) time-step: $h_k = s_{ar}(u_{<k})$. For the CPC task at the segment level we quantize the prediction targets using an online k-means vector quantizer [van den Oord et al., 2017] $q$, which transforms the sequence of candidate units into a sequence of quantized targets $u_1^q, \ldots, u_{T'}^q$: $u_k^q = q(u_k)$. Quantizing the targets has been shown to improve the quality of speech representations in contrastive prediction-based self-supervised learning [Baevski et al., 2020, Nguyen et al., 2022], perhaps by contributing to the prior of discreteness we wish to impose on the high-level features.

## 2.1 Training objective

The objective function capturing our strategy of modeling the signal simultaneously at the level of high resolution low-level continuous features and variable-rate high-level discrete units can be simply defined as the sum of CPC losses $\mathcal{L}_L$ and $\mathcal{L}_H$ operating at both levels. Because the hard decisions of the boundary predictor in the downsampling module are not differentiable, we model the boundary detector as a stochastic reinforcement learning policy. Its loss $\mathcal{L}_\pi$ minimizes the expected value of the high-level loss $\mathcal{L}_H$, whose gradient can be estimated through policy gradient algorithms [Williams, 1992, Sutton et al., 1999]. Finally, the model also learns to optimize the loss $\mathcal{L}_Q$ of the k-means quantizer in the high-level module, and a regularization term $\mathcal{L}_{\bar{l}}$ for the boundary predictor, which reduces the search space of the policy by imposing a prior on the expected average sampling rate of discrete units. The total loss is the sum of all described terms $\mathcal{L} = \mathcal{L}_L + \mathcal{L}_H + \mathcal{L}_Q + \mathcal{L}_\pi + \mathcal{L}_{\bar{l}}$.

**Low-level CPC** The low-level feature extractor is trained to minimize the standard CPC loss

$$\mathcal{L}_L = -\sum_t \sum_{n=1}^N \frac{\exp\left(p_n^T z_{t+n}\right)}{\sum_{z_i \in \mathcal{N}} \exp(p_n^T z_i)}, \tag{1}$$

where $p_1, \ldots, p_N$ are the output of $N$ prediction heads that predict the $N$ upcoming encodings conditioned on $c_t$, and $\mathcal{N}$ is the set of negative samples drawn randomly from the set of encodings out of the prediction window.

**High-level CPC** In contrast to the frame-level features in the low-level CPC loss, where the acoustic frames change smoothly over time, at the level of discrete units we expect successive elements to be distinguishable from each other. Therefore, at the high-level we modify the standard CPC loss to incorporate this prior knowledge of contiguous heterogeneity. We do so by sampling the distractors in the contrastive task from representations adjacent to the prediction target

$$\mathcal{L}_H = -\sum_k \sum_{m=1}^M \frac{\exp\left(p_m^T u_{k+m}^q\right)}{\sum_{i \in \{k+m-1, k+m+1\}} \exp(p_m^T u_i^q)}, \tag{2}$$

where $p_1, \ldots, p_M$ are the output of $M$ prediction heads that predict the $M$ upcoming pseudo-unit representations conditioned on $h_k$ .

**Quantizer** To quantize a pseudo-unit representation $u_k$ we choose the $i$-th embedding $e_i$ from a finite size learnable codebook that minimizes the distance to the candidate unit representation

$$u_k^q = e_i : \min_i ||u_k - e_i||. \tag{3}$$

The codebook is trained through the online k-means loss from van den Oord et al. [2017]

$$\mathcal{L}_Q = \sum_k ||\text{sg}(u_k) - u_k^q)|| - \lambda||u_k - \text{sg}(u_k^q))||, \tag{4}$$

where sg is the stop gradient operator and $\lambda$ is an hyperparameter.

**Boundary predictor** It is modeled as a $\theta$-parameterized stochastic policy $\pi_\theta$ that takes as input the sequence of encodings $z$ and defines for each time-step the probability of a unit boundary conditioned on $z$: $\pi_\theta(b_t|z) = p(b_t|z)$. The policy is trained to minimize the expected value of $\mathcal{L}_H$

$$\mathcal{L}_\pi = \mathbb{E}_b[\mathcal{L}_H(b)|z, \theta] = \sum_b \pi_\theta(b|z)\mathcal{L}_H(b). \tag{5}$$

We use REINFORCE [Williams, 1992] to estimate the gradient of $\mathcal{L}_\pi$ with respect to $\theta$ as

$$\nabla_\theta \mathcal{L}_\pi = \mathbb{E}_b\left[\mathcal{L}_H(b)\nabla_\theta \log(\pi_\theta(b))\right], \tag{6}$$

where the expectation can be approximated through Monte-Carlo methods from experience rollouts.

**Average sampling rate regularization** The model of the boundary detector as a stochastic policy allows us to promote an average pseudo-unit length (or sampling rate) in a differentiable way. Let $\bar{l}$ denote the desired average pseudo-unit length, we define an additional regularization term as

$$\mathcal{L}_{\bar{l}} = \left\|\mathbb{E}_{b_t \sim \pi_\theta}\left[\sum\nolimits_{t=1}^{\bar{l}} b_t\right] - 1\right\| = \left\|\left(\sum\nolimits_{t=1}^{\bar{l}} \pi_\theta(b_t)\right) - 1\right\|, \tag{7}$$

with the right hand side resulting from assuming $b_t$ as a Bernoulli variable. During training we compute equation 7 on randomly sampled segments from all the possible segments of length $\bar{l}$ in a batch. This prior reduces the difficulty of the policy optimization as it restricts the search space to policies that produce segmentations matching the average expected unit length.

## 3    Experimental setup

The experiments were performed on the train-clean-100 subset of LibriSpeech (Panayotov et al. [2015], CC BY 4.0) using the aligned phone labels provided by van den Oord et al. [2018]. To evaluate low-level frame representations, we measured phone prediction accuracy of a linear classifier trained on frozen features. Additionally, we report phone accuracy of the transcriptions obtained from the sequence of representations. It was computed with a 1-layer bidirectional LSTM network with 256 hidden units, followed by a 1D convolution with 256 filters, kernel width 8 and stride of 4, and a CTC loss [Graves et al., 2006]. We modified the loss to forbid emitting the blank token, since as we evaluate on English data only we assume no repetition in successive phones. High-level representations have a lower variable resolution, therefore we evaluated them only using the CTC transcription task. To prevent the case when the predicted transcribed sequence is shorter than the ground truth sequence, we upsample high-level representations according to the length of the segments before compression.

The model is trained to extract segments which on average match the duration of phones, therefore we also assess unit discovery capabilities. We evaluate frame-level representations on the ABX task from the ZeroSpeech 2021 competition [Nguyen et al., 2020], in which the objective is to discriminate between samples that differ on a single unit. We compute the ABX score on the dev-clean set. Furthermore, we report the agreement between segment boundaries learned by the downsampling strategy and the boundaries of human-defined information-bearing units, such as phones. We evaluate phone segmentation performance of the predicted pseudo-unit boundaries through precision, recall, F1-score and over-segmentation robust R-value [Räsänen et al., 2009] with a 20 ms tolerance against phone boundaries obtained from the aligned frame labels. We also evaluate boundary prediction on a processed version of the TIMIT dataset [Garofolo et al., 1993] in which non-speech events have been trimmed to a maximum of 20 ms. We evaluate The setup of our boundary detection evaluation was chosen for comparability with recent studies on unsupervised phone boundary detection [Kreuk et al., 2020, Bhati et al., 2021, Cuervo et al., 2022, Kamper, 2022].

### 3.1    Models

Our model reads single channel raw waveforms sampled at 16kHz, chunked into sequences of 20480 samples (1.28 sec). At the low-level model, the encoder $g_{enc}$ applies five 1D convolutions with internal dimension 256 and filter widths (10; 8; 4; 4; 4). Convolutions are followed by channel-wise magnitude normalization and ReLU activations. Convolutions are strided by (5; 4; 2; 2; 2), resulting in a 160-fold rate reduction, yielding 256-dimensional frame representations extracted every 10ms. The context-building model $g_{ar}$ is a two-layer LSTM network [Hochreiter and Schmidhuber, 1997] with 256 units. At the high-level, $s_{enc}$ is a network with two fully connected hidden layers of 256 units and ReLU activations. $s_{ar}$ is a single layer LSTM network with 256 units. We use $N=12$ and $M=2$ predictions, and 128 and 1 negative samples, for the CPC losses at low and high level models respectively. Each prediction head accesses all past contexts through a single auto-regressive transformer layer [Vaswani et al., 2017] with 8 scaled dot-product attention heads with internal dimension 2048 and dropout [Srivastava et al., 2014] with $p = 0.1$. The target quantizer $q$ uses a codebook of 512 embeddings each of dimension 256, and we use the standard literature value of $\lambda = 0.25$ in equation 4. The boundary predictor is a single layer bidirectional transformer network with 8 scaled dot-product attention heads with internal dimension 2048. We set the expected average unit duration $\bar{l} = 8$ in equation 7 to roughly match the average phone duration measured on the 100 hour subset of the LibriSpeech dataset (7.58 frames).

We compare our model to several baselines. Regarding models with single-level feature extraction: the CPC implementation from [Rivière et al., 2020] which we use as our frame-level model, ACPC [Chorowski et al., 2021] which improved on vanilla CPC by promoting temporally-stable features, and the CPC optimized for phone boundary detection by Kreuk et al. [2020].

Table 1: Frame-level representations evaluation. Linear frame-wise accuracy and CTC phone accuracy on the test split of LibriSpeech train clean 100, and ABX within and across speakers on the ZeroSpeech 2021 dev-clean set. Our model results in frame representations with better linear separability of phones than the ones obtained from a model trained with supervised downsampling. Overall the model improves phone discriminability.

| Architecture | Model | Frame accuracy $\uparrow$ | Phone accuracy $\uparrow$ | ABX within $\downarrow$ | ABX across $\downarrow$ |
|---|---|---|---|---|---|
| Single level | CPC [Rivière et al., 2020] | 67.50 | 83.20 | 6.68 | 8.39 |
| | ACPC [Chorowski et al., 2021] | 68.60 | 83.33 | 5.37 | 7.09 |
| Multi-level | Two-level CPC no downsampling | 67.49 | 83.38 | 6.66 | 8.34 |
| | SCPC [Bhati et al., 2021] | 43.79 | 68.38 | 20.18 | 16.26 |
| | Two-level CPC w. downsampling | 67.92 | 83.39 | 6.66 | 8.32 |
| | mACPC [Cuervo et al., 2022] | 70.25 | 83.35 | 5.13 | 6.84 |
| | Ours | **72.57** | **83.95** | **5.08** | **6.72** |
| | Downsampling (supervised) | 71.01 | 84.70 | 5.07 | 6.68 |

For multi-level modeling we compare: a stack of two CPC modules both using the architecture from Rivière et al. [2020] in which the output $z$ vectors of the low-level model are used as input for the high-level model (two-level CPC) with no downsampling; a stack of two CPC modules in which the high-level model downsamples the outputs of the low-level model at a fixed rate (two-level CPC with downsampling) using a 1D conv with dim 256, filter width 9 and stride of 9, resulting in a sampling rate close to the average phone rate of LibriSpeech clean 100; SCPC [Bhati et al., 2021] using a boundary predictor based on detecting peaks of feature pairwise dissimilarities; mACPC which improves upon SCPC on representation learning by adding long-term prediction at both levels [Cuervo et al., 2022]. Finally, we add a supervised topline in which the boundary-predictor is forced to return ground-truth boundary locations and otherwise is equal to our two-level CPC model.

For the model of Kreuk et al. [2020] we base our implementation in the authors' implementation obtaining consistent results with the ones reported. For SCPC and mACPC we use the implementations provided in Cuervo et al. [2022]. We make our code available at `https://github.com/chorowski-lab/hCPC`.

For single-level models we train until convergence for 50 epochs on the train split of LibriSpeech clean 100. Multi-level models are trained for 50 additional epochs after appending to the pre-trained frame-level modules the high-level network and downsampler, where applicable. We obtained similar results using a training schedule with a weighted loss in which during the first epochs the frame-level loss outweighs the high-level one. However, this method required careful tuning, so we defaulted to the more stable frame-level pre-training.

All models are trained using a batch size of 64, the Adam optimizer [Kingma and Ba, 2015] with a learning rate of 0.0002, and an initial warm-up phase where the learning rate is increased linearly during the first 10 epochs. For the quantization module we follow the setup from Łańcucki et al. [2020]. Training our model takes around 13 hours on 2x Nvidia RTX 3080 GPUs. Evaluation models are trained for 10 epochs using the same hyperparameters, with the exception of the warm-up learning rate schedule.

## 4 Results

**Low-level representations** In the frame-level evaluation (Table 1), the models which perform content-dependent downsampling have the highest phone classification accuracies and ABX scores. Conversely, two-level CPC baselines with constant or no downsampling show little to no improvements over single-level models. The proposed model achieves higher frame accuracy than the model with downsampling according to phone labels, hinting at a possible misalignment between the objectives of matching human-defined phone boundaries and disentanglement of representations. Finally, we observe poor accuracy of SCPC despite having two levels. This is consistent with [Bhati et al., 2022, Cuervo et al., 2022], and caused by the absence of context modeling in the frame-level module

Table 2: High-level representations evaluation. Average sampling rate and CTC phone accuracy on the test split of LibriSpeech train clean 100. Our model gives the best results in phone accuracy and has the lowest average sampling rate among unsupervised methods with variable downsampling.

| Downsampling | Model | Avg. sampling rate (Hz) ↓ | Phone accuracy ↑ |
|---|---|---|---|
| None | Two-level CPC no downsampling | 100 | 83.41 |
| Constant | Two-level CPC with downsampling | 10.94 | 67.75 |
| Variable | SCPC [Bhati et al., 2021] | 15.91 | 55.49 |
| | mACPC [Cuervo et al., 2022] | 14.47 | 69.66 |
| | Ours | **12.32** | **78.93** |
| | Downsampling (supervised) | 10.87 | 85.74 |

of SCPC, which optimizes for boundary prediction (see below), but cripples the ability to capture phone class information.

Transcription performance measured by phone accuracy shows a smaller difference between unsupervised models, likely due to preservation of similar amount of phonetic information in the representation, which a non-linear classifier network is able to access. Incorporating top-down information about the structure of the signal might promote disentanglement of low-level representations, therefore making them more amenable to linear discrimination.

**High-level representations** Table 2 shows the results of the evaluations on high-level representations. Our model performs best among unsupervised models with variable downsampling, both in terms of compression rate and phone transcription accuracy. The model with no downsampling obtains the best performance on phone transcription among unsupervised models, showing that all downsampling strategies result in lossy compression. On the other hand, the model which downsamples the signal according to phone labels aligned with supervision exceeds the performance of not downsampling, proving that a better compression is possible and can be beneficial in such unsupervised transcription task (c.f. discussion in sec. 5.2).

**Matching phone boundaries** Boundary detection results are presented in Table 3. The boundaries of units predicted by our model often match with human annotated phone boundaries. On the LibriSpeech dataset our models attain the highest precision among tested models, at the expense of lower recall. Through a visual inspection of the boundaries made by other models we learned that the remaining models tend to predict false positive boundaries inside non-speech segments, which explains their poor precision. This is caused by segmentation criteria of detecting peaks of dissimilarity between adjacent representations. Non-speech segments of audio show little variation, and noise can easily trigger the peak detector. We confirm this by performing boundary detection on TIMIT without non-speech events, in which dissimilarity-based segmentation models give the best overall performance. We conclude that due to a different training objective, our model might be more robust to the quality of recordings.

## 5 Discussion

### 5.1 On the relative importance of the priors

Our model is based on several priors about the high-level structure of the speech signal, implemented through architectural choices. This begs to question the relative impact of them on the results. Table 1 and Table 2 already show the importance of the priors implied by multi-level modeling and variable-rate downsampling. To complement these results we provide in Table 4 an ablation study on the priors imposed to the high-level features, such as average downsampling rate through policy regularization, and the prior of discreteness using adjacent negative sampling and target quantization. We present the results of the ablation in terms of linear frame-wise phone classification accuracy, in which our model improved the most with respect to the baselines, and phone-boundary detection performance, to illustrate the effect of the priors on the resulting downsampling strategies.

Table 3: Phone boundary detection results on the test split of LibriSpeech train clean 100 (top) and TIMIT test split (bottom). For comparability with Kreuk et al. [2020], Bhati et al. [2021], Cuervo et al. [2022] non speech events were removed from TIMIT. Our model produces segmentations competitive with the state-of-the-art, while being robust to non-speech events.

| Dataset | Architecture | Model | Precision | Recall | F1 | R-val |
|---|---|---|---|---|---|---|
| LibriSpeech clean 100 | Single level | [Kreuk et al., 2020] | 61.12 | 82.53 | 70.23 | 61.87 |
| | Multi-level | mACPC [Cuervo et al., 2022] | 59.15 | **83.17** | 69.13 | 57.71 |
| | | SCPC [Bhati et al., 2021] | 64.05 | 83.11 | 72.35 | 66.40 |
| | | Ours | **79.94** | 77.92 | **78.91** | **81.98** |
| TIMIT (non-speech removed) | Single level | [Kreuk et al., 2020] | 84.80 | **85.77** | 85.27 | 87.35 |
| | Multi-level | mACPC [Cuervo et al., 2022] | 84.63 | 84.79 | 84.70 | 86.86 |
| | | SCPC [Bhati et al., 2021] | **85.31** | 85.36 | **85.31** | **87.38** |
| | | Ours | 80.08 | 81.40 | 80.73 | 83.50 |

Table 4: Ablation on the priors of average sampling rate (policy regularization) and discreteness (adjacent negative sampling and target quantization) in terms of frame-wise linear phone classification accuracy and phone boundary detection R-score on the test split of LibriSpeech train clean 100.

| Policy regularization | Adjacent negative sampling | Target quantization | Frame accuracy | R-val |
|---|---|---|---|---|
| ✓ | ✗ | ✗ | 64.13 | 21.10 |
| ✗ | ✓ | ✗ | 66.87 | 0.0 |
| ✓ | ✓ | ✗ | 72.52 | 81.09 |
| ✓ | ✓ | ✓ | **72.57** | **81.98** |

Adjacent negative sampling, through which we implement the prior of unit distinguishability, appears essential. Without it, the boundary predictor converged to predict a roughly constant probability, sampling the boundaries during training roughly every four frames. In early experiments, which we add to the supplemental material, we observed that using the standard strategy for negative sampling at the high level to train the boundary detector resulted in rewarding over-segmentation. The model turned the high-level task into the easier problem of short term prediction at the level of smooth acoustic features. When coupled with the average sampling rate constraint, this seemed to result in a random production of segments, and frame accuracy plunged even below single-level models.

Policy regularization in terms of the expected average segment length is another crucial element. Without it, frame accuracy dropped close to that of a single-level model. Segmentations produced at the early stages of training were unreasonable, therefore not producing any informative experience rollout for the policy gradient estimator. Eventually, the policy would collapse to a local minimum of not predicting any boundaries at all. This could be addressed by promoting exploration or through a different parametrization of the actions space. Note that the average segment length prior can be seen both as a feature and a limitation. It allows to control the time scale at which the signal is modeled. However it implies some degree of supervision in an ideally unsupervised method, particularly in our experimental setup we used the training data to estimate it. In other applications and domains defining a characteristic sampling rate might be difficult.

Finally, target quantization seems to contribute the least to the accuracy. However, we observed that it tended to accelerate training convergence. Early experiments available in the supplemental material also show that target quantization provides a stronger supervision signal for the reinforcement learning policy by more strongly penalizing unreasonable segmentations.

## 5.2 On human defined units vs. discovered pseudo-units for representation quality

Results at both low and high level display a non-intuitive misalignment between matching human-defined unit (phone) boundaries and improving representations. At frame-level, it seems that the objective of maximizing for discrete predictable structure at the high-level, which not necessarily means pushing towards phonetic structure, results in more useful top-down feedback than down-sampling to match phonetic information content. At high-level however, our optimization objective

results in phonetic information loss, as we were not able to match the phone accuracy of the model with downsampling according to phonetic units. It could be possible that there are patterns between n-grams of phones, which map to a single high-level unit, and facilitate the predictive task more than at the level of individual phones. Yet, by being many-to-one mappings, they could make it impossible to retrieve the original constituents after compression. We consider that an interesting objective for future work is characterizing the patterns of the discovered pseudo-units sequences, and perhaps investigate their correlation with models of human speech perception.

## 6   Related Work

Perhaps the most widely used variable-length feature discovery approach in deep learning is the extraction of subword units from text [Schuster and Nakajima, 2012, Sennrich et al., 2016, Kudo and Richardson, 2018]. These approaches rely on exact symbol matches to detect segments. They serve as inspiration but are not directly applicable to learning non-uniform segmentations of dense representations. Segmental unit-discovery in Kamper et al. [2017] chunked the training corpus into segments that were then clustered into a small vocabulary. We use similar assumptions in our system: every frame belongs to a segment and the high-level representations are trained to match categories.

The discovery of acoustic units in unlabeled data has been approached with non-parametric Bayesian inference on engineered features (e.g. MFCCs). Lee and Glass [2012] proposed a Gibbs sampling inference on a Dirichlet Process HMM while Ondel et al. [2016] derived a Variational Bayes scheme. A similar segmental HMM model has been coupled with a deep-learning based autoencoder in the HMM Variational Autoencoder Ebbers et al. [2017]. Variable-length unit discovery in an end-to-end deep learning setting was recently proposed by Chorowski et al. [2019] who modified the VQ-VAE quantization objective to promote token stability between neighboring frames and Dieleman et al. [2021] who used run-length encoding of VQ-VAE tokens. These approaches are similar in spirit to the dissimilarity-based peak detector used in SCPC [Bhati et al., 2021]: they detect segments by merging frames with similar representations. In contrast, we learn a generic boundary detection which is trained using top-down supervision from the upper-level CPC loss operating on segments.

Hierarchical modeling of data is sometimes performed in the context of generative models. VQ-VAE 2 [Razavi et al., 2019] trained a language model on discrete tokens and generated output images in two steps: it first sampled sequence of discrete tokens form the language model, then sampled the image conditioned on them. Similarly Lakhotia et al. [2021] trained two-level language models on speech. However, in these approaches the second-level models did not influence the learning of the low-level features. We have demonstrated the benefits of using the high-level loss to tune the low-level representation.

Finally, recently proposed systems that train speech recognizers on unpaired speech and transcriptions [Yeh et al., 2018, Baevski et al., 2021] typically require a step which approximately detects segments of similar frames and tries to pair transcriptions with speech using these noisy segments. These methods could benefit from using our proposed segmentation and high-level features.

## 7   Conclusions

We have successfully demonstrated the benefits of training a two-level CPC system which extracts frame-synchronous low level features using a first-level CPC loss, and then discovers a variable-rate segmentation using supervision from a second high level CPC criterion. A key component of our system is a segment boundary predictor trained using REINFORCE to directly optimize the second-level CPC objective. This leads to the discovery of discernible and predictable second-level segments which coincide with a ground-truth phonetic segmentation. Moreover, the extra supervision from the second level yields improvements to the quality of the frame-level representations, nearly matching that of a supervised topline using ground-truth phonetic boundaries.

## Work contribution of authors

The idea of training two-layer CPC systems was discussed between the authors while Jan Chorowski was still with the University of Wrocław. All experimental work was done by Santiago Cuervo

under supervision of Adrian Łańcucki, Ricard Marxer and Paweł Rychlikowski. The manuscript was written by Santiago Cuervo, Adrian Łańcucki, Ricard Marxer and Paweł Rychlikowski.

## Acknowledgements

The authors thank the Polish National Science Center for funding under the OPUS-18 2019/35/B/ST6/04379 grant and the PlGrid consortium for computational resources. We also thank the French National Research Agency for their support through the ANR-20-CE23-0012-01 (MIM) grant.

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
