# Supplemental material -
# Variable-rate hierarchical CPC
# leads to acoustic unit discovery in speech

**Santiago Cuervo**[1,2]      **Adrian Łańcucki**[3]      **Ricard Marxer**[2]

**Paweł Rychlikowski**[1]      **Jan Chorowski**[4]

[1] University of Wrocław, Poland
[2] Université de Toulon, Aix Marseille Univ, CNRS, LIS, France
[3] NVIDIA, Poland
[4] Pathway, France

## 1 Experiments on the high-level CPC loss for unit discovery

In order to assess how the high-level CPC loss would promote unit-discovery in the variable-rate representations, we performed several experiments in which we observed the evolution of the loss during training for multiple downsampling strategies. The plots below illustrate the obtained results. Intuitively, we would like our discovered units to roughly correspond to phones (human defined units). Since downsampling is trained through reinforcement learning to minimize the high-level loss, we want the lowest loss in the plots to correspond to something similar to downsampling as per phone boundaries.

In Figure 1 we show the learning curves when using the standard CPC loss at the high-level. We see that the lowest curves correspond to segmentations with over-sampling. This happens because the model turns the high-level task into the easier problem of short term prediction at the level of smooth acoustic features.

In Figure 2 we show the case when we apply negative sampling from adjacent representations. In this case the lowest loss corresponds to downsampling as per phone boundaries. This shows that the prior of contiguous dissimilarity effectively breaks the incentive of the model to capture smooth continuous patterns, and encourages it to model the signal at the level of discrete units.

In Figure 3 we show the case when we apply negative sampling from adjacent representations and target quantization. We see that target quantization more strongly penalizes over-sampling and uniform rate downsampling, as the curves for these cases become more distinguisable from the curve for the case of downsampling as per phone boundaries. In practice, this results in faster and more stable convergence, as it provides a stronger supervision signal for the downsampling policy.

## 2 Qualitative analysis of detected boundaries

In this section we present some visual results obtained from analyzing the learned segmentations. Figure 4 shows multiple examples of predicted segmentations by a fully trained model, and illustrates the significant agreement between predicted pseudo-unit boundaries and phone boundaries. Figure 5 depicts the distribution of the segment lengths obtained by the model and the distribution of phone

36th Conference on Neural Information Processing Systems (NeurIPS 2022).

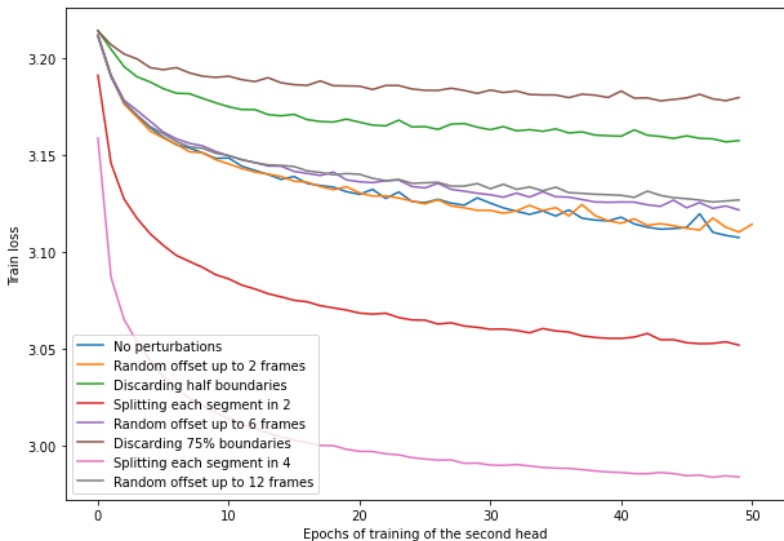

Figure 1: Evolution of the high level loss across 50 epochs of training for different downsampling schemes when using the standard CPC loss.

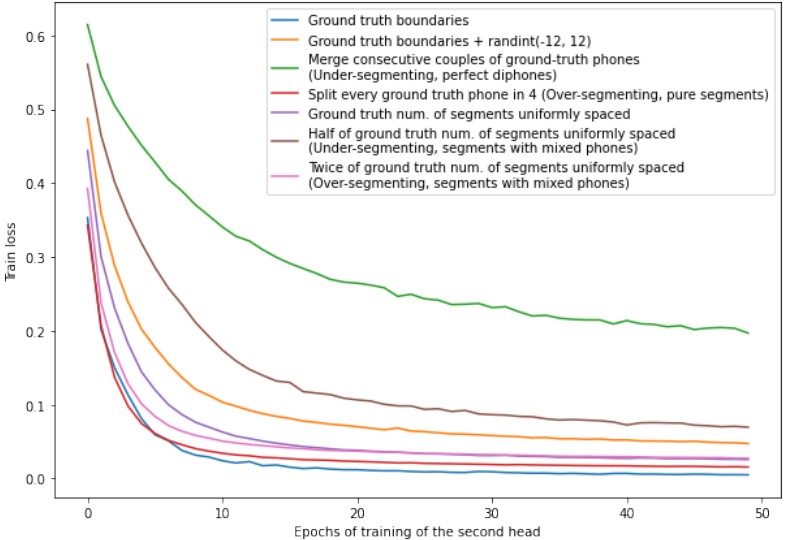

Figure 2: Evolution of the high level loss across 50 epochs of training for different downsampling schemes when using the CPC loss with adjacent negative sampling.

lengths, both calculated on the test split of LibriSpeech train-clean-100 [Panayotov et al., 2015] in units of frames.

Similarly to Bhati et al. [2022], we tried to determine to which degree the class of the phones between which a boundary is located affects the capability of the model to detect it. We perform an experiment in which for every consecutive pair of phones which occurs on the dataset we count how many times the model correctly predicts the boundary between them. To facilitate visualization, we present the results not between individual phones, but between phone families. Figure 6 shows the percentage of correct predictions between the different phone families on the test split of LibriSpeech train-clean-100 for our model with learned segmentation (top) and for SCPC [Bhati et al., 2021], a model with segmentation based on peaks of cosine dissimilarity. There does not seem to b!e an agreement between the kind of mistakes that both segmentation schemes make. As pointed in Bhati et al. [2022], their model struggles with boundaries between phones in which the transition is smooth and gradual. This can be explained by the segmentation method, which relies on abrupt transitions by design. The errors in our method might be more influenced by linguistic patterns, eg. commonly

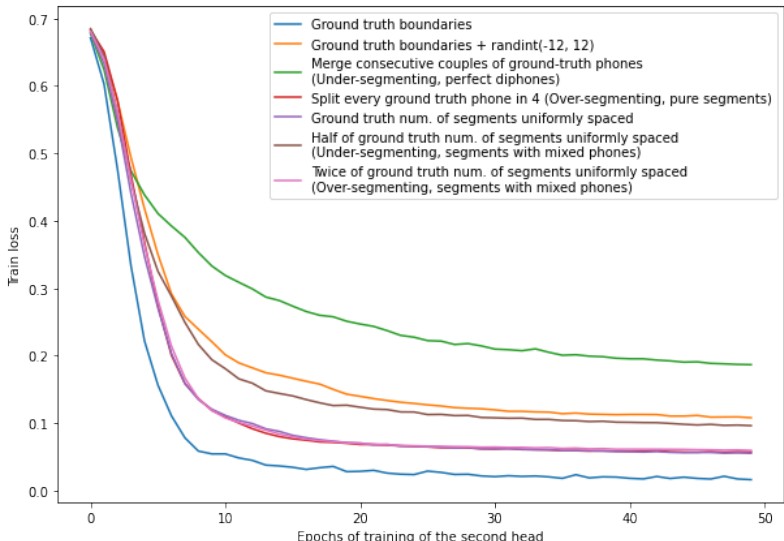

Figure 3: Evolution of the high level loss across 50 epochs of training for different downsampling schemes when using the CPC loss with adjacent negative sampling and target quantization.

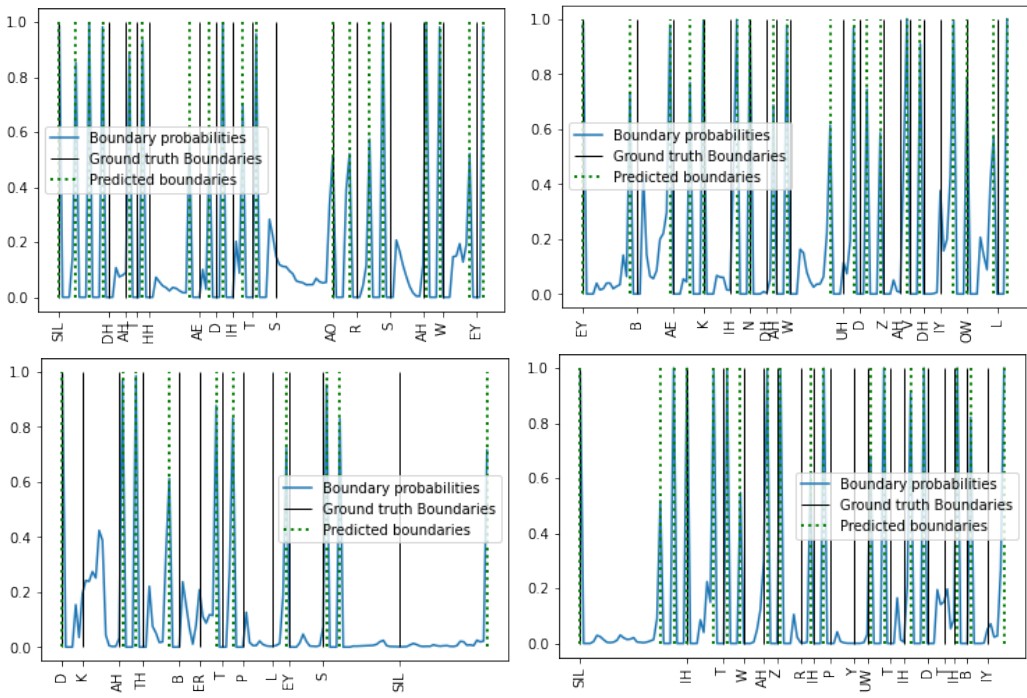

Figure 4: Some predicted segmentations.

ocurring bigrams of phones getting assigned to a single unit. We plan to further characterize the discovered pseudo-units to better understand such differences.

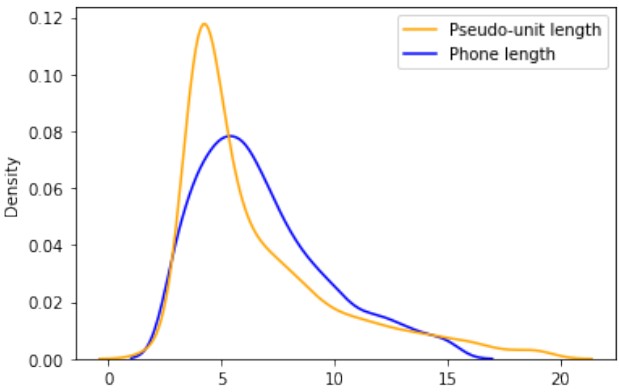

Figure 5: Comparison of distribution of discovered pseudo-unit lengths and phone lengths in number of frames.

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

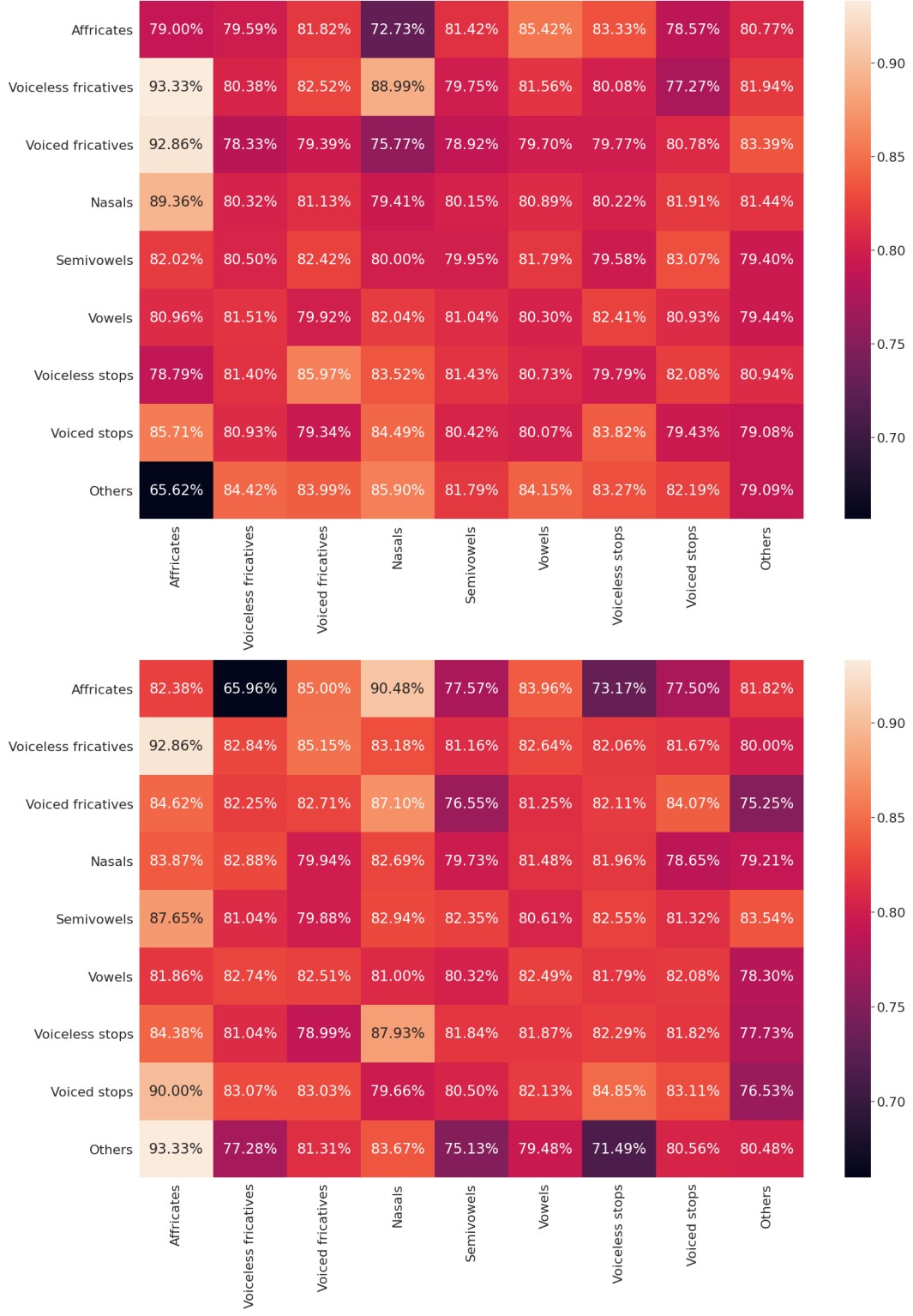

Figure 6: Recall of boundaries between phone families for our learnable boundary detector (top) and a boundary detector based on peaks of cosine dissimilarity (bottom).