# OpenReview forum: "Variable-rate hierarchical CPC leads to acoustic unit discovery in speech"
_NeurIPS.cc/2022/Conference — NeurIPS 2022 Accept_

### Official Review · Reviewer_EQcC · 2022-07-10

**Rating:** 6
**Confidence:** 4
**Soundness:** 3 good
**Presentation:** 4 excellent
**Contribution:** 3 good

**Summary:**

This paper studies unsupervised acoustic unit discovery using a hierarchical CPC training criterion. In particular, the CPC loss is computed at two levels: frame-level and segment-level. Segmentations are produced by a jointly trained boundary detector, which can be understood as a policy network that minimizes the top-level CPC loss optimized using the REINFORCE algorithm.

The authors further introduce two crucial designs, without which the secondary CPC does not improve the performance from single-level CPC. The first design is how distractors of the top-level CPC are selected. The authors show in ablation studies that it is important to select distractors from the adjacent frames of the target one to encourage contrastiveness. The second is to inject prior knowledge of average phone duration in the boundary prediction module.

The proposed model achieves better frame-level phone accuracy, segment-level phone accuracy, and phone segmentation compared to several baselines.


**Questions:**

1. Why do the authors modify CTC to not predict blank symbols for evaluation if phone accuracy is computed at the sequence level?
2. Following the previous question, do phone accuracy in Table 1 and Table 2 both represent the same metric and are both computed at the sequence level? If they are computed at the frame level / segment level instead, how are the segment level ground truth phones determined if a segment contains frames from more than one phones?

See weaknesses for other questions


**Limitations:**

Yes, the authors have addressed limitations.

**Strengths And Weaknesses:**

Strengths:
1. The paper is easy to follow, with all the design choices well motivated and clearly described.
2. The experiment section also contains a good set of ablation studies and baselines to justify the importance of design choices.
3. Strong results compared to the baselines

Weaknesses
1. The authors could have included ABX results on the frame-level feature, segment-level features, and quantized frame/segment features. This would be more directly related to “unit discovery.” The evaluation included in the paper only concerns boundary detection.
2. More ablation studies can be added.
    - It is shown that selecting adjacent representations of the target one is beneficial for high-level CPC. Would it also be helpful for training low-level CPC?
    - In line 202, the author states “Multi-level models are trained for 50 additional epochs after appending to the pre-trained frame-level modules the high-level network and downsampler, where applicable.” How important is it to pre-train the frame-level modules? It would be nice to also show the results of training everything from scratch.
    - Add boundary detection results to Table 4
3. (minor) The following statement is wrong: “In a similar spirit Lakhotia et al. [2021] trained language models on raw speech. However, in these approaches the second-level models still operated at a frequency synchronous with the input one.” In fact, Lakhotia et al. [2021] and Kharitonov et al. [2021] both use deduplicated unit sequences ([1, 1, 2, 3, 3, 3] -> [1, 2, 3]) for training LMs, so the LM operate on the segment level instead of on the frame level.

Kharitonov, Eugene, et al. "Text-free prosody-aware generative spoken language modeling." arXiv preprint arXiv:2109.03264 (2021).

---

> ### Author Response · Authors · 2022-08-02
> **Response to reviewer EQcC**
>
> Thank you for the thoughtful review and valuable feedback!
>
> > The authors could have included ABX results (...)
>
> Following your comment, we have computed ABX scores on the ZeroSpeech 2021 dev-clean set for our frame-level representations, obtaining results that further confirm the strength of our model on phone discrimination:
>
> | Model | ABX within  | ABX across |
> |-|-|-|
> | CPC [Rivière et al., 2020] | 6.68 | 8.39 |
> | ACPC [Chorowski et al., 2021] | 5.37 | 7.09 |
> | SCPC [Bhati et al., 2021] | 20.18 | 16.26 |
> | mACPC [Cuervo et al., 2021]  | 5.13 | 6.84 |
> | Ours  | **5.08** | **6.72** |
>
>
> We have included these results in the revised supplementary material. We will further elaborate on ABX scores for the camera-ready version, including scores on segment-level features.
>
> > More ablation studies can be added.
> > 1. It is shown that selecting adjacent representations of the target one is beneficial for high-level CPC. Would it also be helpful for training low-level CPC?
> > 2. In line 202, the author states (...) How important is it to pre-train the frame-level modules? It would be nice to also show the results of training everything from scratch.
> > 3. Add boundary detection results to Table 4
>
> 1. In our experiments with adjacent negative sampling at frame level, the resulting representations were not useful for phone classification nor boundary prediction, as the negatives are too strongly correlated with the positives. The motivation for adjacent negative sampling at the high-level CPC is to encode the prior of distinguishability of adjacent discrete units. Low-level features change smoothly, making this sampling strategy unsuitable.
>
> 2. Without pre-training of the low-level model, the high-level loss tends to dominate in early training, resulting in the boundary predictor converging to predict a roughly constant probability. In this case the random segmentations seem to prevent the low-level module (and therefore the high-level module) from learning useful representations. We have managed to obtain similar results to the ones described in the paper using a loss-schedule in which during the first epochs the low-level loss outweighs the high-level one. However, this method requires careful tuning, so we defaulted to the more stable low-level pre-training.
>
> We will mention these findings in the camera-ready version of the paper.
>
> 3. We have added boundary detection R-score to Table 4 in the revised version of the paper. We include the results here for easy access:
>
> | Policy regularization | Adjacent negative sampling | Target quantization | Frame accuracy (\%) | R-val (\%) |
> |-|-|-|-|-|
> | ✓ | X | X | 64.13 | 21.10 |
> | X | ✓ | X | 66.87 | 0.0 |
> | ✓ | ✓ | X | 72.52 | 81.09 |
> | ✓ | ✓ | ✓ | **72.57** | **81.98** |
>
> > (minor) The following statement is wrong: “In a similar spirit Lakhotia et al. [2021] trained language models on raw speech. (...)
>
> Thank you for spotting our error! We unfortunately missed this information from the cited paper and blindly assumed that since the model featured resynthesis, no downsampling was performed, we are sorry for this mistake.
>
> The revised sentence is:
> In a similar spirit Lakhotia et al. [2021] trained two-level language models on
> speech. However, in these approaches the second-level models did not influence the learning of the low-level representations.
>
> From our experiments with wav2vec 2.0, the quantized codes changes ca 2.5 times more often than phoneme, thus another mechanism which enforces longer segmentations is needed. From several ones we have tried (also including changes to the discretizer to change the cluster id less often), the proposed two-level CPC worked best.
>
> > Why do the authors modify CTC to not predict blank symbols (...)
>
> CTC tends to predict mostly blank symbols, with occasional peaks on non-blank symbols. We initially removed blank symbols to force an alignment between the sequences of phones in the reference and the representations. Furthermore, we assume no repetition in successive phones both in the method and the evaluation since we test on English only data. We have further explained this in the revised version.
> We will redo the phone accuracy experiments with blank symbols and update the tables for the camera-ready version if significant differences are observed.
>
> > Following the previous question, do phone accuracy in Table 1 and Table 2 both represent the same metric (...)
>
> They both represent the same metric and they are both computed at sequence level. At segment level, the sequence is upsampled according to the length of the segments before compression. We have added clarifications in the revised version of the paper.

---

> > ### Comment · Reviewer_EQcC · 2022-08-09
> > **Thank you**
> >
> > I thank the authors for the response and it has sufficiently addressed my questions.

---

### Official Review · Reviewer_fYLw · 2022-07-10

**Rating:** 6
**Confidence:** 4
**Soundness:** 3 good
**Presentation:** 3 good
**Contribution:** 3 good

**Summary:**

This paper proposed to achieve unsupervised acoustic unit discovery in speech using a specially designed variable-rate hierarchical CPC method. The low-level CPC is applied at a frame-level while the high-level CPC is applied by non-uniformly downsampling the low-level CPC outputs. The non-uniformly downsampling is performed based on high-level unit segmentations estimated by a boundary predictor, which is trained using REINFORCE.

**Questions:**

1. "We consider that an interesting objective for future work is characterizing the patterns of the discovered pseudo-units sequences, and perhaps investigate their correlation with models of human speech perception."

This seems to be a very interesting direction, but it's very possible that human and machines have different solutions. I wonder how do the authors think to make the findings useful rather than "inventing" a different phonology.

In particular, there exists very-well performed algorithms to achieve automatic frame-to-subword-unit alignment without requiring any phone labels. This could easily achieve much better phone accuracy and possibly a better averaged sampling rate.

2. What if some collar (e.g. 1-2 frames) is allowed in the phone boundary detection experiments?

**Ethics Review Area:**

["I don’t know"]

**Limitations:**

1. This research direction is certainly very interesting from a scientific perspective, but I would still hope the authors to discuss more about the potential use of such automatically-discovered acoustic units.

2. Hope to see more statistics and analysis about the automatically discovered high-level acoustic units. It's unclear to me how many of them were found, how well they correspond to the phones, and how about their durations etc.?

3. Acoustic unit generation, clustering and discovery have been long-standing research problems in the speech community. Although the related work tried to cover some recent influential works related in the NLP community, the pioneering and more directly related works from the speech communities are mostly ignored. Actually it would be quite interesting to digging into those works and find the connections and relations. At least, even if the current framework of related work is kept, this paper should be cited as it is the original paper for the widely used wordpiece models:
https://static.googleusercontent.com/media/research.google.com/en//pubs/archive/37842.pdf


**Strengths And Weaknesses:**

Strengths:
The paper is easy to follow. The proposed method is novel and does technically sound from the speech perspective.

Weaknesses:
1. Figure 1 should be redesigned to be more descriptive and better looking. The resolution should be considerably improved and better to use a vector image.
2. The formats of the references are not consistent.
3. The results may not be strong enough to demonstrate the superiority of the proposed method. For instance:
(1) In Table 1, the improvement in phone accuracy is small over existing method. I am not sure how sensible it is to evaluate the frame accuracy of phonemes here (as the phoneme boundaries are known to be noisy).
(2) In Table 2, although the proposed method achieved the best phone accuracy among the downsampling-based methods, the result is still quite a bit worse than the no-downsampling method.
(3) In Table 3, the phone boundary detection results are quite a lot behind previous numbers.

---

> ### Author Response · Authors · 2022-08-02
> **Response to reviewer fYLw**
>
> Thank you for the thoughtful review and valuable feedback!
>
> Regarding technical comments, Figure 1 has been redrawn in the revised version of the paper, and the entries in the references section have been updated and made coherent.
>
> > The results may not be strong enough to demonstrate the superiority of the proposed method. (...)
>
> (1) The increase in frame accuracy (over 2%) seems to be significant with respect to other methods to be attributed to noise. Furthermore, the increase should not be attributed to shifts in the representations towards a direction favoring an increase in frame accuracy, as we outperform the work of Cuervo et al. [2022], in which representation shifting is explicitly punished in a way that was shown to favor agreement with the used alignments. Therefore,  we believe our results on frame-accuracy show a clear improvement in linear separability of the representations.
>
> (2) Although there is a significant performance gap between low-level and high-level representations, we believe that the results of table 2 are significant in the sense of showing that our learnable downsampling results in more efficient compression than other strategies. Adding an auxiliary reconstruction task could allow more control over the loss of information in the downsampling bottleneck, and could improve the performance of high-level features.
>
> (3) Our method was not explicitly designed for phone segmentation, as it is the case of the methods to which we compare in Table 3. With the segmentations results we meant to show that the variable-rate segmentations learned by our model have a meaningful interpretation, as they showed a significant agreement with human-defined segmentations, namely phone boundaries.
>
> > (...) I wonder how do the authors think to make the findings useful rather than "inventing" a different phonology. (...)
>
> Our work, and future steps have to be considered from the vantage point of unsupervised representation learning with no transcriptions, so subword units cannot be applied. We do not attempt to beat supervised alignment mechanisms (e.g. having access to word-level transcripts), but to investigate if imposing the segmentality constraint on an unsupervised learner is beneficial.
>
> We plan to study how the representations learned from the proposed unsupervised approach relate to human listening results, such as those found in consistent confusion corpora or other intelligibility experiments. We will focus on their relative predictive capacity when compared to other self-supervised speech representation approaches rather than on a functional comparison with the human speech perception mechanisms. This should provide us insights on the effects on the inductive biases associated to each method on the quality of the representations.
>
> Seeing where human and algorithmic units diverge might lead to a refined design of models - for instance equipped with more precise priors.
>
> > What if some collar (e.g. 1-2 frames) is allowed in the phone boundary detection experiments?
>
> We allow for a tolerance of 2 frames, following the setup of the works with which we compare to in phone boundary detection experiments. We have stated this explicitly in the revised version of the paper.
>
> > This research direction is certainly very interesting (...) discuss more about the potential use of such automatically-discovered acoustic units.
>
> The primary objective of acoustic unit discovery is unsupervised speech recognition, especially devised for low-resource languages. Further applications could extend to other domains: analysis of speech of people with speaking disabilities, language acquisition process in toddlers. Distant signal processing domains could include analysis of sounds of animals, land and marine, or insects. Unit discovery in spontaneous speech could be used for identification and synthesis of non-speech events, such as coughing, laughing, sneezing, etc. for more engaging speech synthesis. Lastly, variable-length segmentation of 2-d and 3-d signals could find applications in object detection for computer vision or biomedical imaging.
>
> > Hope to see more statistics and analysis about the automatically discovered high-level acoustic units. (...)
>
> In the paper we evaluate the discovered boundaries of potential acoustic units, and how well the learned representations of speech are disentangled. Please note that boundary prediction implies duration prediction. We do not attempt extraction of complete acoustic unit inventories, which is still considered to be a difficult task in a purely unsupervised regime.
>
> > (...) the pioneering and more directly related works from the speech communities are mostly ignored. (...)
>
> A reference to the paper which introduced WordPieceModel was indeed omitted, and we have added it to the revised version of our paper. In the related work section, we now refer to pioneering approaches to AUD, such as non-parametric Bayesian models, and autoencoder-based HMMs.

---

> > ### Comment · Reviewer_fYLw · 2022-08-09
> > **Thanks for the response**
> >
> > Thanks for the careful responses to my comments. The authors have resolved most of my concerns and I would increase my rating.

---

### Official Review · Reviewer_wUr5 · 2022-07-11

**Rating:** 5
**Confidence:** 4
**Soundness:** 2 fair
**Presentation:** 2 fair
**Contribution:** 3 good

**Summary:**

This paper proposes a new method to segment pseudo-phoneme boundaries by combining multi-level CPC and boundary predictor agent. The reward for the boundary predictor agent is the high-level CPC objetive, thereby producing sparse boundary flag. The learned representations show nice performance on phoneme prediction and boundary detection tasks.

**Questions:**

- line 92, conditioned on **c** → doesn’t make sense judging from Figure 1. I guess it’s **z** not **c**?
- line 92, and and ?
- line 132, again, c → z
- what are **p**s in equation (1) and (2)? what nn architecture was used to produce this?
- why does the policy trained to minimize the expected value of L_H makes the agent predict the phoneme boundaries? Is there any guarantee for this?

**Limitations:**

It is written in the disccussion part.

**Strengths And Weaknesses:**

strengths

- The idea of training boundary predictor using reinforcemnent learning is a clever idea.

weakness

- The performance of the training algorithm significantly reduces when there is no average frame sample rate regularization (Eq. (7)). This implicitly demonstrates that the agent fail to find the boundaries without the help of regulaizration and show that the reward function for the agent may not be an appropriate one.
- Maybe the proposed training scheme cannot really gurantee to find the optimal boundary if there is no endeavour to reconstuct the original signal.
- It'd have been nicer if there were some qualitative analysis.

---

> ### Author Response · Authors · 2022-08-02
> **Response to reviewer wUr5**
>
> Thank you for the thoughtful review and valuable feedback!
>
> Mistakes in writing spotted on lines 92 and 132 have been corrected in the revised version of the paper, along with Figure 1.
>
> > The performance of the training algorithm significantly reduces when there is no average frame sample rate regularization (Eq. (7)). This implicitly demonstrates that the agent fail to find the boundaries without the help of regulaizration and show that the reward function for the agent may not be an appropriate one.
>
> The experiments described in the supplement show that the reward function does promote phone boundary detection, also over the under-segmentation, to which the model eventually collapses without average sampling rate regularization. Therefore, we have reasons to believe that the collapse of the policy without regularization might not be caused by an ill-suited reward function, but rather issues with policy optimization, e.g., lack of exploration, or high variance in the REINFORCE estimator. Currently we are working on exploration strategies and alternative parametrizations of the policy which could provide reasonable segmentations without average sampling rate regularization. We also believe that policy gradient estimators with less variance, e.g., using an advantage function, could render policy regularization no longer necessary.
>
> > Maybe the proposed training scheme cannot really gurantee to find the optimal boundary if there is no endeavour to reconstuct the original signal.
>
> A reconstruction loss would indeed provide an easier way of controlling information loss after compression at the high-level. However, speech is highly redundant. Considering a recording with long silence regions or prolonged vowels, a reconstruction of the original might over-emphasize easier segments of the signal.
>
> Experimental results of our method show that the boundaries tend to be correct. In addition, we have presented all boundary detection experiments, and did not hold back any failed attempts.
>
> > It'd have been nicer if there were some qualitative analysis.
>
> We have added some qualitative analysis of the discovered boundaries and a comparison of the distribution of segment lengths and phoneme length to the supplementary material.
>
> > What are ps in equation (1) and (2)? what nn architecture was used to produce this?
>
> Those are the outputs of functions whose input are the context vectors (see line 119). In line 174 we specify that the architecture used to implement each of those functions is an auto-regressive transformer, following the CPC setup from Rivière et al. [2020]. We have added a short explanation to make it clearer in the revised version.
>
> > Why does the policy trained to minimize the expected value of L_H makes the agent predict the phoneme boundaries? Is there any guarantee for this?
>
> The task defined by $L_H$ encourages the model to find a compression of the speech signal into a sequence in which 1) Its elements are discrete, 2) are unevenly distributed in time, and 3) exhibit structure useful for predicting the future given the context. By setting the desired average sampling rate to match the average phone sampling rate, we therefore expected the model to discover something similar to phone sequences, as they fulfill 1-3). Our results show that this expectation is met to some degree, but the mismatches also suggest that there are other sequences with the desired properties, and as such there is no guarantee of finding phone boundaries. It is worth mentioning that although phone boundaries were the expected outcome, it was not our goal in itself. We intended to show that accounting for the inherent language-related structure in speech (phonemes, words, etc.), which implies the existence of underlying sequences with properties 1-3), would improve representations. The fact that this resulted in segmentations with a significant agreement with phone boundaries was a welcomed result that further validated our approach.

---

### Comment · Area_Chair_6hdm · 2022-08-07
**reviewer-author discussion**

Dear reviewers,

The authors have posted up their rebuttals.  If you have not done so, please engage with the authors to discuss your feedback or any further concerns.

Thanks,

AC

---

### Meta-Review · Area_Chair_6hdm · 2022-08-23

**Recommendation:** Accept
**Confidence:** Certain

**Metareview:**

In this paper the authors propose to use multiple levels of contrastive predictive coding (CPC) in self-supervised learning to discover acoustic units.  The lower level of CPC learning is a conventional one while the upper level of CPC is carried out based on non-uniform downsampling with boundary prediction. To facilitate the training, the authors introduce various loss functions for hierarchical CPCs, quantization, boundary predictor and average sampling rate regularization.  The authors show that the acoustic units discovered by this framework give better phone accuracies in both frame and segment levels compared to existing models.   The idea of variable-rate hierarchical CPC learning is interesting and experimental results are supportive.  The paper is well written and easy to follow.  Most of the raised concerns by the reviewers have been addressed in the rebuttal.  To further improve the paper,  the authors may want to show the effectiveness of this model to generate better acoustic representations in more downstream speech domain and tasks.

**Award:**

No

---

### Decision · Program_Chairs · 2022-09-14

Accept